health and disease and epidemiology/ mathematical modelling

vaccination COVID-19, health cost, population pyramid, vaccination campaign

**Author for correspondence:**
Daniel Gros
e-mail: danielg@ceps.eu

# When to end a lock down? How fast must vaccination campaigns proceed in order to keep health costs in check?

Claudius Gros[1], Thomas Czypionka[2,3] and Daniel Gros[4]

[1]Institute for Theoretical Physics, Goethe University Frankfurt, Frankfurt am Main, Germany
[2]Institute for Advanced Studies, Vienna, Austria
[3]Department of Health Policy, London School of Economics and Political Science, London, England
[4]CEPS (Centre for European Policy Studies), Brussels, Belgium

TC, 0000-0002-3381-1075; DG, 0000-0002-2911-8235

We propose a simple rule of thumb for countries which have embarked on a vaccination campaign while still facing the need to keep non-pharmaceutical interventions (NPI) in place because of the ongoing spread of SARS-CoV-2. If the aim is to keep the death rate from increasing, NPIs can be loosened when it is possible to vaccinate more than twice the growth rate of new cases. If the aim is to keep the pressure on hospitals under control, the vaccination rate has to be about four times higher. These simple rules can be derived from the observation that the risk of death or a severe course requiring hospitalization from a COVID-19 infection increases exponentially with age and that the sizes of age cohorts decrease linearly at the top of the population pyramid. Protecting the over 60-year-olds, which constitute approximately one-quarter of the population in Europe (and most OECD countries), reduces the potential loss of life by 95 percent.

## 1. Introduction

Infection with the SARS-CoV-2 virus represents a serious health risk. A number of studies have by now established that this risk increases exponentially with age (see [1] for a survey and metastudy). This applies both to the risk of dying and to the risk of a severe course of the illness requiring hospitalization [2].

Vaccination campaigns have therefore prioritized the elderly (along with health workers). Countries with active outbreaks have in general introduced NPIs to restrict mobility and reduce the number of personal contacts [3]. However, NPIs have a high social and economic cost. This cost arises both through the limitations to economic activity [4] and the health costs from

those catching the virus and becoming infected [5]. It is thus key to understand when NPIs can be lifted safely. This has led to a race between mass scale vaccination campaigns and the disease, which continues to spread and evolve. Dye *et al.* [6] provide an analysis of the scale and the spread of the disease in 2020. Zheng *et al.* [7] provide a mathematical model. Agosto *et al.* [8] provide an overview of the statistical models used to forecast the spread of the disease. A notable application of statistical modelling to the European experience is provided by [9] and (using a Poisson process) [10]. Moreover, new variants have appeared [11], with new and potentially more infectious strands [12].

The ultimate aim of vaccination campaigns is to achieve 'herd' or population immunity, i.e. a state in which the share of the population that has been immunized by infection or vaccination brings down the effective reproduction number $R_{eff}$ below unity without further NPIs needed [13]. However, vaccination takes time, mostly due to the difficulties in ramping up the production of vaccines [14], but also in part also due to bottlenecks in distribution and implementation. A core issue for policy makers therefore is the point during the vaccination campaign at which NPIs can be lifted.

We provide a general rule to answer this question. Our proposed rule is based only on observable data and is the result of combining two simple relationships

— The age-dependency of the case fatality and the hospitalization rate, which has been established to increase approximately exponentially with age [1].
— The population structure for the elderly, which is to first order linearly at the top, which means that the size of age cohorts increases gradually top-down from the maximum age (about 100 years) [15].

We concentrate on observable outcomes like death or hospitalization because their number constitutes a key determinant for the imposition of lock downs [16] and other NPIs, which come with severe economic and social costs in terms of lost output and employment [17] as well as indirect impacts on health and well-being [18].

The combination of these two elements leads to a simple rule of thumb to recognize the 'sweet point' at which NPIs can be loosened while still 'flattening the curve'. We then add a third element, namely

— The functional dependency of daily vaccinations rates, which are observed to increase in most countries approximately linearly over time.

This third element allows one to predict the stage in the vaccination process at which the 'sweet point' will be reached. We concentrate throughout mostly on the case of Europe, because this is the region in which a resurgence of infections coincided with an, initially at least, sluggish vaccination campaign.

## 2. Modelling framework

### 2.1. Age-dependency of the health risks of COVID-19 infections

We concentrate on two observable health risks: death and hospitalization.

It has been widely documented that the risk of dying from a COVID-19 infection rises strongly with age. A meta study suggests an exponential relationship [1], which can be parameterized as

$$\text{IFR} \approx 0.01 e^{-7.529 + 0.121 * a} \sim e^{a/a_0}, \quad a_0 = 8.26, \tag{2.1}$$

where $a \in [0, 100]$ is the age cohort. The infection fatality rate $\text{IFR} \in [0, 1]$ is very high for $a = 100$, namely $\text{IFR}(100) = 0.93$.

The constant $a_0$ denotes the half-life age difference in terms of mortality. To be more precise, for an age difference of 8.26 years the risk increases by a factor of $e = 2.78$. The risk doubles for an age difference of 5.7 years.[1]

Limiting the strain on health systems has been another constant concern of policy makers. The risks of requiring hospitalization is also age specific. Statistics from the US CDC show that people aged 85 years and older face a risk of hospitalization from COVID-19 infections about 95 times higher than that of the age group 5–17. This translates also into a doubling age difference of about 16 years ( $= (85 - 11)/\ln(95)$ ), which we denote as $a_{0h}$ about double of that of the fatality risk.

---

[1]Statistics from the CDC of the US show that the over 85 years old have a COVID-19 mortality rate 7900 times higher than that of the age group 5–17. This translates also into a doubling age difference of about 8 years ( $= (85 - 11)/\ln(7900)$ ) [2].

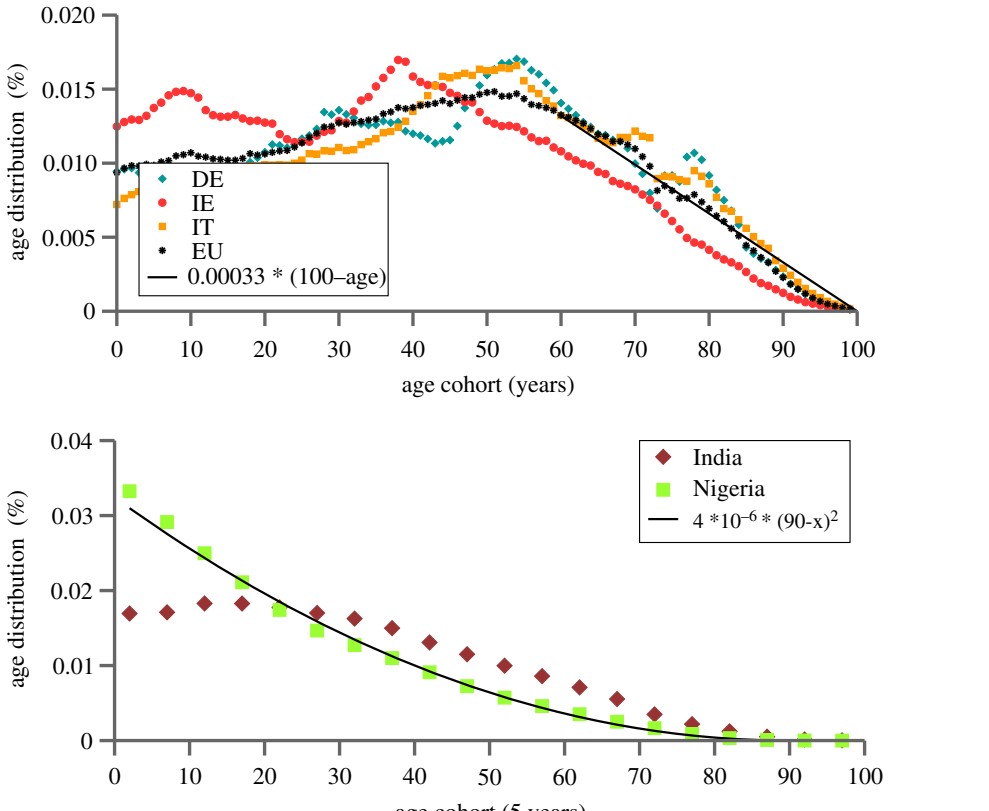

**Figure 1.** Age distributions. (*a*) Fractional yearly age cohorts for Germany (DE), Ireland (IE), Italy (IT) and EU27 (EU). Also shown is a linear interpolation for the EU data (black line), for ages 60 and above. Data from [19]. (*b*) 5-year age cohorts for India and Nigeria. A quadratic interpolation to the data for Nigeria is also provided.

In the following, we will concentrate on decision rules designed to limit fatalities, but we will also refer to consequences of this somewhat different age dependency when the aim is to limit the pressure on the health system.

## 2.2. Age pyramid

We present the age pyramid for a range of selected countries in figure 1, where the age cohorts are given as a share of the entire population. One notices that the age pyramid closes generically quadratically at the very top. For EU countries, the relevant range for the quadratic dependency is however restricted, applying only to ages around 85 and above.

Most COVID-19 vaccination campaigns started by following top-down strategies to varying degrees [20]. For the time being, we concentrate on EU countries, for which the respective age pyramids can be approximated linearly, as shown in figure 1. This approximation is intended as an overall fit to ages 60 and above. The case of countries with younger populations, like India and Nigeria, will be treated later [21].

In the following, we set the maximum age to zero, counting down from 100. The actual age is then $100 - x$. The age density, denoted by $\rho(x)$, varies by cohorts. Countries with population pyramids closing linearly at the top are described by

$$\rho(x) = px, \quad \int_0^{\delta A} \rho(x)\,\mathrm{d}x = v, \quad v = \frac{p(\delta A)^2}{2}, \tag{2.2}$$

where $v$ is the number of people (relative to the total population) vaccinated top-down to an age difference $\delta A$.

Fitting to aggregate European demographic data yields $p \approx 0.00033$, as shown in figure 1, with little difference across the major EU member countries. This value for $p$ refers to the case that $\rho(x)$ is measured

relative to the total population, in terms of percentiles of cohorts by year. For example, with this value of $\rho(x)$ the share of the 70-year-olds in the total population is equal to about 0.01 =0.00033*30.

The linear approximation holds well down to 60 years. Below this age, the size of the cohort no longer increases (and even falls in some countries, like Italy). Here we concentrate on the age cohorts from 60 years up, which are the ones subject to the highest mortality risk, constituting the largest proportion of the overall loss of life. The case of Germany illustrates this proposition. Taking into account the combined effect of (2.1) and the age distribution, as presented in figure 1, one finds that about 1.5 million people above the age of 60 die in the hypothetical scenario that the entire population would be eventually infected with SARS-CoV-2. By contrast, the fatality count would include only 75 thousand below 60, i.e. by a factor of 20. We thus feel justified concentrating our analysis on the age cohorts above 60, for which the population pyramid is approximately linear.

People over 60 years of age account for about 26 percent of the total population of the EU, with their shares ranging from 20 percent in the case of Ireland to 29 per cent in the case of Italy. This implies that vaccinating about one-fourth of the population will avoid 95 per cent of fatalities (19/20). This calculation based solely on age obviously represents an approximation. Due to vaccine hesitancy, non-responders and people with contraindications to the vaccine, the uptake among the elderly could be less than 100 percent. But these factors are also present among all age groups, reducing thus the overall benefit from a vaccination campaign, but not necessarily the advantage of age-sensitive vaccination. Vaccine hesitancy is in particular likely to be lower among the elderly, implying that the share of the benefits from offering vaccination to the elderly first might be even higher than the 95% suggested on demographic considerations alone. A factor suggesting otherwise may however be 'long COVID' [22].

There is also evidence that immunity wanes more quickly at higher ages [23], implying that the re-infection risk is higher for the elderly. This effect, however, plays out on a time scale beyond that of most current vaccination campaigns.

# 3. Flattening the health cost curve

Putting the two basic elements—the exponential age-dependency of the case fatality rate, the linear functionality of population pyramid—together, we proceed to calculate the 'sweet spot' at which the vaccination campaign can stabilize health costs.

We concentrate on the *growth* of fatalities because this is the key concern for policy makers. But we also show that our approach can easily be used to address the generic concern 'flatten the curve', i.e. to prevent an explosive increase in hospitalizations which could overwhelm health systems.

Throughout this section we take the growth of infections as given. In §6, we discuss how the results are modified by the feedback between vaccination and infections.

## 3.1. Vaccination and infection fatality rate

We assume that full vaccination provides a high level of protection against severe illness and death, as confirmed not only by trial data [24] but also by real-world application [25,26]. For our model, we furthermore assume that vaccination is allocated strictly by age, starting with the oldest. In practice, the situation is more complicated. Firstly, because a substantial fraction of the available vaccine is reserved for potential spreaders in most countries [20], independent of their age. Secondly, one needs to distinguish between people having received one or two shots. Both effects could be incorporated into the framework developed here. In order to clarify the mechanisms at work, we study in the following the idealized situation that 'vaccinated' implies full protection.

# 4. Medical balancing condition

In a first step, we derive a medical balancing condition that is generically valid, viz for all types of population pyramids. In a second step, we will make use of the fact that population pyramids may often be well approximated linearly at the top. As before, we do not work with the nominal age $a \in [0, 100]$, but with the relative age $x$, as measured top-down, $a = 100 - x$, denoting by $\rho(x)$ the respective population density. The fraction $v$ of people vaccinated top-down to an age difference $\delta A$ is

then

$$v = \int_0^{\delta A} \rho(x) \, dx, \quad \text{and} \quad \frac{dv}{dt} = \rho(\delta A) \frac{d(\delta A)}{dt}, \tag{4.1}$$

which included also a differential relation with regard to the vaccination speed $\dot{v}$. The above expression reduces to $v = p(\delta A)^2/2$ for a population pyramid closing linearly at the top, viz for $\rho(x) = px$.

A key variable for policy makers is the medical load, i.e. the number of patients requiring hospitalization and maybe even intensive care or who die. We start by considering the mortality risk. The medical load or cost for society at any given state of the vaccination campaign can then be expressed as the number of people who risk dying each time period:

$$C^{\mathrm{med}} = \left[ \int_{\delta A}^{\infty} \rho(x) \, e^{-x/a_0} \, dx \right] I(t), \tag{4.2}$$

where is was assumed that fully vaccinated people will not get seriously ill anymore. Daily COVID-19 fatalities do not increase when $C^{\mathrm{med}}$ remains stable, viz when the balancing condition $dC^{\mathrm{med}}/dt = 0$ holds. Setting the time derivative of the right-hand side of (4.2) to zero leads to the condition that

$$\frac{d\delta A}{dt} [\rho(\delta A) \, e^{-\delta A/a_0}] = \left[ \int_{\delta A}^{\infty} \rho(x) \, e^{-x/a_0} \, dx \right] \frac{\dot{I}}{I}. \tag{4.3}$$

With the expression for $\dot{v}$ given in (4.1), the left-hand side of (4.3) reduces to $\dot{v} \exp(-\delta A/a_0)$. We then have

$$\dot{v} = e^{\delta A/a_0} \left[ \int_{\delta A}^{\infty} \rho(x) \, e^{-x/a_0} \, dx \right] \frac{\dot{I}}{I}. \tag{4.4}$$

This expression is valid for all population densities $\rho(x)$, in particular also for the tabulated population density of a given country.

## 4.1. Linear population pyramid

Evaluating the integral in (4.4) for the linear case, when $\rho(x) = px$, one has

$$p \int_{\delta A}^{\infty} x \, e^{-x/a_0} \, dx = -(pa_0^2) \left[ 1 + \frac{x}{a_0} \right] e^{-x/a_0} \Big|_{\delta A}^{\infty}$$

$$= (pa_0^2) \left[ 1 + \frac{\delta A}{a_0} \right] e^{-\delta A/a_0}. \tag{4.5}$$

With

$$v_0 = \frac{a_0^2 p}{2}, \quad v = \frac{p \delta A^2}{2}, \quad \frac{\delta A}{a_0} = \sqrt{\frac{v}{v_0}} \tag{4.6}$$

one then finds that (4.4) reduces to

$$\frac{\dot{v}}{v_0} = 2 \left( 1 + \sqrt{\frac{v}{v_0}} \right) \frac{\dot{I}}{I}, \tag{4.7}$$

where $\dot{I}/I$ is the (relative) increase of the incidence.

# 5. Consequences of the balancing condition

## 5.1. The sweet spot for an old country

When the population pyramid is linear at the top (as it is typically for ageing societies), the balancing condition (4.7) leads to a simple rule of thumb, given that $v_0$ is equal to about 0.01, or one percent:

> For every proportional increase $\dot{I}/I$ of the incidence, one needs to vaccinate an additional percentage of *at least* twice that amount in order to outrun the virus.

This lower bound holds for $v \to 0$, becoming larger when vaccination progresses. The balancing condition thus provides a simple decision rule based on observable data (infections and vaccinations). Once this 'sweet spot' has been reached, NPIs can be gradually loosened without risking an increase in fatalities.

## 5.2. The sweet spot: hospitalizations

The balancing conditions derived above describe the point at which the number of fatalities (per unit of time) stops increasing. One key parameter used to derive these results was $a_0$, which denotes the half live age difference in terms of mortality. For the risk of hospitalization this parameter, $a_{0h}$, is about twice as high, but the functional form remains the same.

For a linear age pyramid this implies that the evolution of hospitalizations can be described by the same balancing condition,

$$\frac{\dot{v}}{v_{0h}} = 2\left(1 + \sqrt{\frac{v}{v_{0h}}}\right)\frac{\dot{I}}{I}, \tag{5.1}$$

where the subscript $_h$ indicates that the value is calculated with $a_{0h}$ instead of $a_0$. This changes two of the three relationships in equation (4.6):

$$v_{0h} = \frac{a_{0h}^2 p}{2}, \quad v = \frac{p\delta A^2}{2}, \quad \frac{\delta A}{a_{0h}} = \sqrt{\frac{v}{v_{0h}}} \tag{5.2}$$

The (relative) increase of the incidence, $\dot{I}/I$ and $v$ are not affected.

Comparing (5.2) and (4.6) shows that $v_{0h}/v_0$ is equal to $a_{0h}/a_0$. It was documented above that the mean doubling age for hospitalizations is about two times larger than that for deaths (16 instead of 8 years). This implies that $v_{0h}$ is about four times larger than $v_0$. It follows that at low vaccination rates ($v$ small) the balancing condition is four times more stringent if the aim is to keep hospitalizations from increasing than if the aim is to keep fatalities in check, i.e. the decision rule would become:

> For every proportional increase $\dot{I}/I$ of the incidence, one needs to vaccinate an additional percentage of *at least* eight times that amount in order to keep the pressure on hospitals constant.

A policy that takes into account the need to limit the pressure on health systems would thus loosen NPIs much later than a policy which concentrates only on fatalities. The difference is a factor of four at low vaccination rates, but it diminishes as a higher proportion is vaccinated (as $v$ increases) because the higher value of $a_{0h}$ enters the denominator of the second term in the brackets on the right-hand side of equation (5.1).

## 5.3. The sweet spot for a young country

As documented below, younger countries have a top of the population pyramid which closes quadratically. In this case, the size of the cohorts counting from age 100 down can be described by

$$\rho(x) = qx^2, \quad v = \frac{q\delta A^3}{3}, \quad \frac{\delta A}{a_0} = \left(\frac{3v}{a_0^3 q}\right)^{1/3}, \quad v_1 \equiv \frac{a_0^3 q}{3} \tag{5.3}$$

and

$$q\int_{\delta A}^{\infty} x^2 e^{-x/a_0}\, dx = -(qa_0^3)\left[2 + \frac{2x}{a_0} + \frac{x^2}{a_0^2}\right]e^{-x/a_0}\Big|_{\delta A}^{\infty}, \tag{5.4}$$

one obtains the balancing condition

$$\frac{\dot{v}}{v_1} = 3\left[2 + 2\left(\frac{v}{v_1}\right)^{1/3} + \left(\frac{v}{v_1}\right)^{2/3}\right]\frac{\dot{I}}{I}. \tag{5.5}$$

A fit to the population pyramid of Nigeria yields $q = 4 \times 10^{-6}$, viz $v_1 = a_0^3 q/3 = 0.0002$, see (2.1) and figure 1. The reference fraction of vaccinated is hence exceedingly small, $v_1 \sim 0.02\%$.

A generalization to the case of a quadratic population pyramid, as given by (5.5), yields a factor $6v_1/v_0 \approx 0.12$ in the limit $v \to 0$, instead of two (given that $v_0$ corresponds to about one percent). At the start, vaccination campaigns have hence strong control capabilities in countries with young populations, viz when the population pyramid is quadratic at the top. This advantage decreases rapidly, however, with the progress of vaccination. To be concrete, dividing equation (4.7) by (5.5) one obtains a factor of 1.7 at $v = 0.1$, when ten percent of the population has been vaccinated top-down.

## 5.4. Connection with the evolution of the pandemic

The rate of increase in the number of infected will be affected by the fraction of the population already vaccinated. The evidence regarding the impact of vaccines on the spread of infections is less clear than the impact of vaccination on the risk of death or a severe course [27]. Some results point to a reduction in transmission of 40 percent [28] whereas others report a much higher impact among the vulnerable population in long-term care facilities [29]. Any impact of vaccination on infectiousness would not change the balancing condition (4.7), which applies in general. Moreover, with vaccination reducing infectiousness, whatever the size of the impact, the growth of the disease spread slows down, ceteris paribus, (to a lower value for $\dot{I}/I$), making it easier to reach the point where the curve 'flattens'.

The relative increase $\dot{I}/I$ appearing on the right-hand side of (4.7) could be evaluated using a dynamic epidemiological model. For the SIR model [30], the simplest case, we have $\dot{I}/I = gS - \lambda$, where $g$ and $\lambda$ are the reproduction and the recovery rates, respectively. The reservoir of susceptibles $S$ decreases with the fraction of the population that is immunized, either because of having been infected, or due to being vaccinated. We leave this approach for further studies, concentrating here on data-driven considerations.

The dynamics of the growth of a pandemic is studied in a large body of the literature. One of the first to employ the canonical model was [31]. Agosto *et al.* [9] employ statistical models to predict the contagion curve and the associated reproduction rate using a Poisson process. See also [32] for time-series models and [33] for a survey.

We thus conclude that the position of the 'sweet spot' in terms of observed infections is not changed if one takes into account the impact of vaccinations on the spread of the disease. With continuing vaccinations the sweet spot becomes a turning point because once it has been reached, vaccination rates could plateau, but the condition for deaths to fall would remain fulfilled, as additional vaccinations reduce the growth of infection, $\dot{I}/I$, which is the determining factor on the right-hand side of (4.7).

## 6. Vaccination hesitancy

In deriving (4.4), we assumed full vaccination of everybody aged above $a = 100 - \delta A$. In practice, there will be a certain hesitancy $h(x) \in [0, 1]$, meaning that a fraction $h(x)$ remains non-vaccinated, even after vaccines have become available to the age cohort $x$. The age-specific indicator $h = h(x)$ is used here as a general term, including people that are not willing to be vaccinated, together with the fraction of the population that cannot be vaccinated due to medical contra-indications. People not developing an immune response despite being fully vaccinated are also subsumed under $h(x)$.

In Europe, the fraction $h(x)$ of not vaccinated starts close to zero in most countries for the very old, remaining in general below 20 percent down to age 60.[2] It is straightforward to generalize the derivation of the medical balancing condition to the case $h(x) > 0$. For this the term

$$c_0 I(t) \int_0^{\delta A} h(x)\rho(x) e^{-x/a_0}\, dx, \tag{6.1}$$

needs to be added to the medical cost $C^{\text{med}}$, see (4.2). One finds

$$\dot{v}\left(1 - h(\delta A)\right) = e^{\delta A/a_0}\left[\int_{\delta A}^{\infty} \rho(x)\, e^{-x/a_0}\, dx + \int_0^{\delta A} h(x)\rho(x)\, e^{-x/a_0}\, dx\right]\frac{\dot{I}}{I}, \tag{6.2}$$

which reduces to (4.4) when $h(x) \to 0$. Clearly, it becomes more difficult to retain medical balancing when vaccine hesitancy is large, which is however not the case for the elderly.

For the case of age-independent hesitancy, $h(x) \equiv h$, one can evaluate both the first and the second integral on the right-hand side of (6.2). The latter is

$$hp \int_0^{\delta A} x\, e^{-x/a_0}\, dx = -h(pa_0^2)\left[1 + \frac{x}{a_0}\right] e^{-x/a_0}\Bigg|_0^{\delta A}$$

$$= -h(pa_0^2)\left[1 + \frac{\delta A}{a_0}\right] e^{-\delta A/a_0} + h(pa_0)^2. \tag{6.3}$$

[2]See https://www.ecdc.europa.eu/en/publications-data/data-covid-19-vaccination-eu-eea

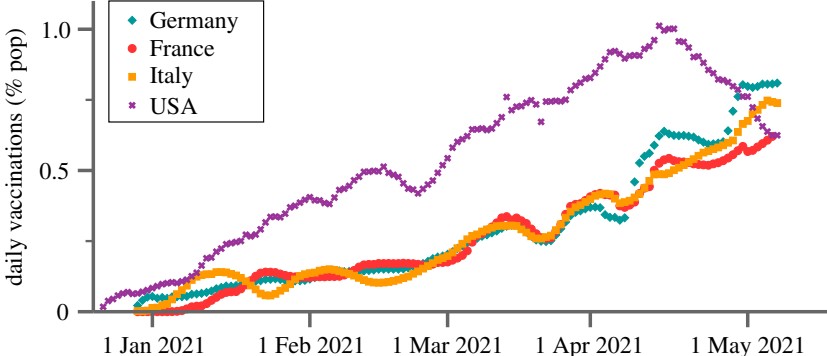

**Figure 2.** Growth of daily vaccination rates. Per million daily vaccinations in Italy, France, Germany and the USA. The growth is roughly linear, modulo in part substantial fluctuations. Data smoothed over 7 days, from [34].

The first term on the right-hand side is $(-h)$ times the obtained for the case without hesitancy. All together we hence have

$$\frac{\dot{v}}{v_0} = 2\left(1 + \sqrt{\frac{v}{v_0}}\right)\frac{\dot{I}}{I} + 2h\frac{e^{\delta A/a_0}}{1-h}\frac{\dot{I}}{I} = 2\left[1 + \sqrt{\frac{v}{v_0}} + h\frac{e^{\delta A/a_0}}{1-h}\right]\frac{\dot{I}}{I}, \tag{6.4}$$

which generalizes (4.7). Vaccine hesitancy thus introduces an explicit age dependency, given that cohorts of age $a$ are defined via $a = 100 - \delta A$. The basic message of the balancing without vaccine hesitancy thus remains, namely that it takes a vaccination campaign that is at least two times quicker than the percentage spread of the disease.

# 7. Evolution of vaccination campaigns

It is not possible to vaccinate the entire population instantly, because vaccines have to be first mass-produced and then distributed. This is illustrated in figure 2, where the daily vaccination rates are shown for a range of selected countries. Daily rates may vary, in particular for smaller countries, when larger batches are delivered from abroad. But for larger entities, like the EU or the USA, the trend is linear. Overall, vaccination rates can be expected to track deliveries, with eventual organizational problems leading only to temporary delays.

Over the course of several months, the daily vaccination rates shown in figure 2 rise roughly linearly during the initial states of the campaign, in line with steadily increasing production capacities.[3] Given these considerations, and the data presented in figure 2, we assume that the number of daily jabs, viz the vaccination rate, increases linearly. The fraction of the population $v$ vaccinated top-down increases then with the square of time $t$,

$$v = v(t) = \frac{1}{2}\left(\frac{t}{t_0}\right)^2 = \frac{p}{2}(\delta A)^2, \quad \delta A = \frac{t}{t_0\sqrt{p}} \equiv a_0 g_v t. \tag{7.1}$$

The parameter $t_0$ denotes the time needed to vaccinate one half of the population. Given a linear increase of cohort sizes with age, one finds that the age of the youngest cohort that can be fully vaccinated, denoted $\delta A$, falls linearly over time, see (2.2). The factor $a_0$ in the last definition is the characteristic age determining the exponential functionality of the IFR, as defined by (2.1).

An order of magnitude estimate for the length of vaccination campaigns, $t_0$, can be evaluated from available data. For example, in Israel it took about 10 weeks (from the beginning of January of 2021 to mid-March 2021) to fully vaccinate half the population, resulting in an estimate of $t_0 = 10$ (weeks). In the EU, only about 5 percent of the population was fully vaccinated over the same period corresponding to an estimate of $t_0 = 10\sqrt{10} \approx 32$ (weeks).[4]

---

[3]This linear ramp-up has been predicted [35]. The reason is that COVID-19 vaccine were ordered ahead of their approval in large batches. But ramping up production capacities implies substantial adjustment costs. Minimizing these adjustment costs, subject to fulfilling the order within a contracted time period, leads to a linear ramping-up of production capacities [14].

[4]The time needed to vaccinate the entire population, i.e. to the point $v = 1$, is equal $\sqrt{2}t_0$. The parameter $t_0$ thus does not denote the full length of the vaccination campaign, but the time needed to vaccinate 50% of the population. At that point, more than 99% of the

The balancing condition (4.7) for the case of linear population density can be used to write the sweet spot at which medical costs are kept from rising, in terms of the two structural parameters $a_0$ and $p$,

$$\frac{t}{t_0} = \left(a_0^2 p + \frac{t}{t_0} a_0 \sqrt{p}\right) t_0 \frac{\dot{I}}{I}, \quad v_0 = \frac{a_0^2 p}{2}, \tag{7.2}$$

see (2.1) and (2.2).[5]

As above, one could substitute the parameter $a_0$ with $a_{0h}$ if the aim is to keep hospitalizations under control. Given that equation (7.2) contains the square of the parameter it follows that it will take about four times as long to reach the sweet spot in terms of hospitalizations than in terms of fatalities.

# 8. Discussion

A key aim for policy makers grappling with a continuing outbreak, even when an increasing proportion of the population is being vaccinated, is to 'flatten the curve', i.e. to keep hospitalizations and fatalities from rising exponentially [37]. We identified a balancing condition, or 'sweet spot', at which the health costs remain constant as the most vulnerable are being vaccinated first. The balancing condition can be interpreted as indicating the path of relaxing NPIs along which infections can still increase, but fatalities or death remain under control.

Our aim is not to provide detailed epidemiological modelling and simulations. Instead, we have shown that three key factors can be combined into a simple formula that determines the impact of vaccination campaigns with regard to the time evolution of medical costs. First, the mortality risk from a COVID-19 infection (and that of hospitalization) increases exponentially with age. Second, the sizes of age cohorts decrease from the top of the population pyramid. Third, vaccination proceeds at an increasing speed.

We find considerable differences across two dimensions: the form of the age pyramid and the measure of health costs: hospitalizations or fatalities.

— Age pyramid: Older countries need to vaccinate more quickly than younger ones. The difference between a typical European country and a high fertility country in Sub-Saharan Africa can be a factor of 6.
— Deaths or hospitalizations: For an old country, aiming at keeping hospitalizations constant entails a requirement of a vaccination rate about four times larger than if the aim is to keep fatalities constant. For a young country, the difference is even larger.

Our discussion has focused on the case of Europe, but the formula we derive holds generally and can take into account the wide differences one observes in the speed of vaccination campaigns and the age structure of the population. Moreover, our approach is general enough to accommodate the emergence of different strains of the virus which might increase contagion and/or the risk of severe courses. For example, the impact of the diffusion of the so-called delta variant, whose diffusion has recently been modelled [38], would show up in a higher rate of growth infections, which would need a correspondingly higher speed of vaccination to offset its higher contagiousness.

## 8.1. Limitations

In our framework, 'vaccinated' implies full immunity, which is attained for most COVID-19 vaccines only after the second jab. A reduced levels of immunity, like 95 percent, is equivalent to an equivalent degree of vaccine hesitancy, which we also discuss. Note that only a proportion of the jabs, and not all, is administered following a strict age criterion [39]. Vaccine hesitancy and other factors, such as waning immunity with age, can reduce the overall effect of vaccination campaigns. Moreover, there are other, less age specific costs of the disease, like 'long COVID' [40]. Incorporating these factors would refine the model, at the same time making it necessary to estimate an increased number of parameters. See in this regard, e.g. [41,42] for recent contributions using state-of-the-art epidemiological models.

We have concentrated on the fact that vaccination basically eliminates the risk of death [43]. Vaccination reduces however also the spread of the virus [44]. This provides an additional element which increases the

---

fatalities can be avoided and NPIs can be lifted. $t_0$ provides thus a good parametrization of the effective length of a vaccination campaign.

[5]We concentrate on the general analytical solution in order to avoid having to make too many specific assumptions about the way the pandemic spreads. For a more detailed, structural approach see [36].

importance of vaccination speed. However, this element becomes significant primarily in later stages of a vaccination campaign, beyond the point when the vulnerable groups have been vaccinated.

Data accessibility. The electronic supplementary material contains the data analysed. Only publicly accessible data have been used and the full data is available in the electronic supplementary material. (Sources: CDC on risk of death and hospitalization, United Nations population data and 'Our World in Data' for vaccinations. Here are the websites: https://www.cdc.gov/coronavirus/2019-ncov/covid-data/investigations-discovery/hospitalizationdeath-by-age.html https://population.un.org/wpp/DataQuery/ https://ourworldindata.org/covid-vaccinations

Competing interests. We declare we have no competing interests.

Funding. This research has been supported by the European Union's Horizon 2020 research and innovation program, PERISCOPE: Pan European Response to the Impacts of COVID-19 and future Pandemics and Epidemics, under grant agreement no. 101016233, H2020-SC1-PHE CORONAVIRUS-2020-2-RTD.

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
