## [Peer Review File · Royal Society Open Science]

Review History

RSOS-211055.R0 (Original submission)

Review form: Reviewer 1

Is the manuscript scientifically sound in its present form?

Yes

Are the interpretations and conclusions justified by the results?

Yes

Is the language acceptable?

Yes

Do you have any ethical concerns with this paper?

No

Have you any concerns about statistical analyses in this paper?

No

Recommendation?

Major revision is needed (please make suggestions in comments)

Comments to the Author(s)

The paper concerns a relevant topics and is sufficiently well written. It lacks context: reference to related literature and comparison of findings. Also it lacks clarity: often the formulae are introduced without a clear explanations and the symbols used in the equations are often not clarified or confused. There are also a number of English typos. The relevance of the results of the paper suggests a revision that amends the pitfalls.

A revision should be conducted along the following lines:

1. More literature references should be included, especially with regards to papers that assess the dynamic of the COVID-19 contagion in Europe should be added, and the results therein made complementary with the findings of the paper. Examples are the papers Agosto, Giudici: Risks, 2020; Agosto, Campmas, Giudici, Renda: Statistics in Medicine, 2021.
2. Each formula should be fully explained, stating exactly what each quantity and symbols represents and possibly adding an interpretational comment and a numerical example.
3. The Authors assume almost full vaccine coverage. As they remark, this is not a realistic assumption, and the coverage does also depends on specific types of vaccines. The authors should extend their work in this direction
4. The incidence rate $I/I(\text{hat})$ is not exogenous but depends on vaccines and NPIs. The authors should extend their model to take this aspect into account. Possibly using a statistical approach to estimate the incidence rate for the aggregate EU countries
5. The paper should be better focused, possibly only on European countries for which more data and evidence is available
6. It is not clear whether and why the authors consider only people aged above 60. This should be clarified. Possibly, younger cohorts should be included in the analysis as further covid variants may change the age incidence distribution

Review form: Reviewer 2

Is the manuscript scientifically sound in its present form?

Yes

Are the interpretations and conclusions justified by the results?

Yes

Is the language acceptable?

Yes

Do you have any ethical concerns with this paper?

No

Have you any concerns about statistical analyses in this paper?

No

Recommendation?

Accept as is

Comments to the Author(s)

This is a very elegant article focussing on some key relationships of highly topical importance, and can be published as is. However, given its probable utility to policy makers, it would gain by a table of indicative results for different permutations of key parameters associated with different countries; this would serve to illustrate some of the key points and relationships.

For the same reason, the discussion could be extended to address explicitly how the results could be affected by the presence of new virus variants.

Decision letter (RSOS-211055.R0)

Dear Dr Gros

The Editors assigned to your paper RSOS-211055 "When to end a lock down? How fast must vaccination campaigns proceed in order to keep health costs in check?*" have now received comments from reviewers and would like you to revise the paper in accordance with the reviewer comments and any comments from the Editors. Please note this decision does not guarantee eventual acceptance.

Please submit your revised manuscript and required files (see below) no later than 21 days from today's (ie 14-Sep-2021) date. Note: the ScholarOne system will 'lock' if submission of the revision is attempted 21 or more days after the deadline. If you do not think you will be able to meet this deadline please contact the editorial office immediately.

on behalf of Professor Chris Budd (Associate Editor) and Nick Pearce (Subject Editor)
 openscience@royalsociety.org

Associate Editor Comments to Author (Professor Chris Budd):

Associate Editor: 1

Comments to the Author:

Both of the reviewers comment positively on the relevance and on the presentation of the paper. One reviewer is positive about the paper. However the other reviewer, whilst liking some aspects of the paper, expresses serious concerns about both the context of the paper, in particular the lack of references. They are also concerned that the models are not well described, and that more details need to be given about the derivation of the various formulae used. These are both serious points which I agree with. The authors should therefore carefully revise their paper along these lines.

Reviewer comments to Author:

Reviewer: 1

Comments to the Author(s)

The paper concerns a relevant topics and is sufficiently well written. It lacks context: reference to related literature and comparison of findings. Also it lacks clarity: often the formulae are introduced without a clear explanations and the symbols used in the equations are often not clarified or confused. There are also a number of English typos. The relevance of the results of the paper suggests a revision that amends the pitfalls.

A revision should be conducted along the following lines:

1. More literature references should be included, especially with regards to papers that assess the dynamic of the COVID-19 contagion in Europe should be added, and the results therein made complementary with the findings of the paper. Examples are the papers Agosto, Giudici: Risks, 2020; Agosto, Campmas, Giudici, Renda: Statistics in Medicine, 2021.
2. Each formula should be fully explained, stating exactly what each quantity and symbols represents and possibly adding an interpretational comment and a numerical example.
3. The Authors assume almost full vaccine coverage. As they remark, this is not a realistic assumption, and the coverage does also depends on specific types of vaccines. The authors should extend their work in this direction
4. The incidence rate $I/I(\text{hat})$ is not exogenous but depends on vaccines and NPIs. The authors should extend their model to take this aspect into account. Possibly using a statistical approach to estimate the incidence rate for the aggregate EU countries
5. The paper should be better focused, possibly only on European countries for which more data and evidence is available

6. It is not clear whether and why the authors consider only people aged above 60. This should be clarified. Possibly, younger cohorts should be included in the analysis as further covid variants may change the age incidence distribution

Reviewer: 2

Comments to the Author(s)

This is a very elegant article focussing on some key relationships of highly topical importance, and can be published as is. However, given its probable utility to policy makers, it would gain by a table of indicative results for different permutations of key parameters associated with different countries; this would serve to illustrate some of the key points and relationships.

For the same reason, the discussion could be extended to address explicitly how the results could be affected by the presence of new virus variants.

===PREPARING YOUR MANUSCRIPT===

===PREPARING YOUR REVISION IN SCHOLARONE===

Author's Response to Decision Letter for (RSOS-211055.R0)

See Appendix A.

RSOS-211055.R1 (Revision)

Review form: Reviewer 1

Is the manuscript scientifically sound in its present form?

Yes

Are the interpretations and conclusions justified by the results?

Yes

Is the language acceptable?

Yes

Do you have any ethical concerns with this paper?

No

Have you any concerns about statistical analyses in this paper?

No

Recommendation?

Accept as is

Comments to the Author(s)

The authors have successfully addressed my remarks. The paper can be accepted

Decision letter (RSOS-211055.R1)

Dear Dr Gros,

It is a pleasure to accept your manuscript entitled "When to end a lock down? How fast must vaccination campaigns proceed in order to keep health costs in check?*" in its current form for publication in Royal Society Open Science. The comments of the reviewer(s) who reviewed your manuscript are included at the foot of this letter.

COVID-19 rapid publication process:

We are taking steps to expedite the publication of research relevant to the pandemic. If you wish, you can opt to have your paper published as soon as it is ready, rather than waiting for it to be published the scheduled Wednesday.

This means your paper will not be included in the weekly media round-up which the Society sends to journalists ahead of publication. However, it will still appear in the COVID-19 Publishing Collection which journalists will be directed to each week (<https://royalsocietypublishing.org/topic/special-collections/novel-coronavirus-outbreak>).

If you wish to have your paper considered for immediate publication, or to discuss further, please notify openscience_proofs@royalsociety.org and press@royalsociety.org when you respond to this email.

on behalf of Professor Chris Budd (Associate Editor) and Nick Pearce (Subject Editor)
openscience@royalsociety.org

Reviewer comments to Author:

Reviewer: 1

Comments to the Author(s)

The authors have successfully addressed my remarks. The paper can be accepted

Appendix A

Template for response to referees for RSOS

When to end a lock down? How fast must vaccination campaigns proceed in order to keep health costs in check?

Associate Editor Comments to Author (Professor Chris Budd):

Associate Editor: 1

Comments to the Author:

Both of the reviewers comment positively on the relevance and on the presentation of the paper. One reviewer is positive about the paper. However, the other reviewer, whilst liking some aspects of the paper, expresses serious concerns about both the context of the paper, in particular the lack of references. They are also concerned that the models are not well described, and that more details need to be given about the derivation of the various formulae used. These are both serious points which I agree with. The authors should therefore carefully revise their paper along these lines.

Reaction of the authors:

We wish to acknowledge these useful comments, which have led us to thoroughly revise the paper. Given the large number of changes we made we have not been able to pinpoint all changes made to follow the comments in this document.

Reviewer comments to Author:

Reviewer: 1

Comments to the Author(s)

The paper concerns a relevant topic and is sufficiently well written. It lacks context: reference to related literature and comparison of findings. Also it lacks clarity: often the formulae are introduced without a clear explanation and the symbols used in the equations are often not clarified or confused. There are also a number of English typos. The relevance of the results of the paper suggests a revision that amends the pitfalls.

A revision should be conducted along the following lines:

1. More literature references should be included, especially with regards to papers that assess the dynamic of the COVID-19 contagion in Europe should be added, and the results therein made complementary with the findings of the paper. Examples are the papers Agosto, Giudici: Risks, 2020; Agosto, Campmas, Giudici, Renda: Statistics in Medicine, 2021.

Reaction of the authors:

Many thanks for these references. We have now acknowledged the link with the literature on the dynamics of contagion in Europe.

2. Each formula should be fully explained, stating exactly what each quantity and symbols represents and possibly adding an interpretational comment and a numerical example.

Reaction of the authors:

We apologize for not providing enough detail. We have now endeavored to provide an interpretational comment and numerical example for each quantity. See for example, after equations 2, 3 and 6.

3. The Authors assume almost full vaccine coverage. As they remark, this is not a realistic assumption, and the coverage does also depends on specific types of vaccines. The authors should extend their work in this direction

Reaction of the authors:

We see the relevance of this comment and have added an entire new section (6) which describes what changes when the vaccine coverage is only partial. Many thanks, this is an important extension.

4. The incidence rate I/\hat{I} is not exogenous but depends on vaccines and NPIs. The authors should extend their model to take this aspect into account. Possibly using a statistical approach to estimate the incidence rate for the aggregate EU countries.

Reaction of the authors:

The main objective of the paper is to provide a simple formula for policy makers to calculate the threshold beyond which lock-downs can be lifted. We thus do not assume that the I/\hat{I} is exogenous, only that NPIs can be lifted/reduced once the speed of vaccination reaches a certain threshold related to any given I/\hat{I} . Vaccination reduces, ceteris paribus, the speed of diffusion of the virus (although the extent to which this is the case is not clear, see the new references in the paper). This implies that if the condition for relaxing NPIs is achieved at any given point in time it will also be satisfied in the following period.

5. The paper should be better focused, possibly only on European countries for which more data and evidence is available.

Reaction of the authors:

Many thanks for this comment, which we accept with pleasure since the authors are particularly interested in Europe. Accordingly, we have concentrated our analysis on the case of an 'old' country, which is the case of EU member states.

6. It is not clear whether and why the authors consider only people aged above 60. This should be clarified. Possibly, younger cohorts should be included in the analysis as further covid variants may change the age incidence distribution.

Reaction of the authors:

In the text we comment in particular on the people above a certain age to illustrate the fact that the main danger of loss of life is concentrated among the elderly and because the age pyramid is linear until this age. However, the formula we derive is valid in general, not only for people above 60 years.

It is true that further variants may change the age incidence distribution. This could be incorporated by changing one parameter in our model. We already deal with two aspects: mortality and hospitalizations which have a different age distribution. Moreover, we have added a comment to take possible changes in the age incidence into account.

Reviewer: 2

Comments to the Author(s)

This is a very elegant article focussing on some key relationships of highly topical importance, and can be published as is. However, given its probable utility to policy makers, it would gain by a table of indicative results for different permutations of key parameters associated with different countries; this would serve to illustrate some of the key points and relationships.

For the same reason, the discussion could be extended to address explicitly how the results could be affected by the presence of new virus variants.

Reaction of the authors:

Many thanks for your appreciation of the paper.

We now comment explicitly on the potential emergence of new virus variants, which might be characterized by a changed age incidence distribution. This would necessitate changing only one parameter in our model. We already deal with two aspects: mortality and hospitalizations which have a different age distribution.

===PREPARING YOUR MANUSCRIPT===

- one version identifying all the changes that have been made (for instance, in coloured highlight, in bold text, or tracked changes);
- a 'clean' version of the new manuscript that incorporates the changes made, but does not highlight them. This version will be used for typesetting if your manuscript is accepted.